# Novel Loss-of-Function Variants in *CDC14A* are Associated with Recessive Sensorineural Hearing Loss in Iranian and Pakistani Patients

**DOI:** 10.3390/ijms21010311

**Published:** 2020-01-02

**Authors:** Julia Doll, Susanne Kolb, Linda Schnapp, Aboulfazl Rad, Franz Rüschendorf, Imran Khan, Abolfazl Adli, Atefeh Hasanzadeh, Daniel Liedtke, Sabine Knaup, Michaela AH Hofrichter, Tobias Müller, Marcus Dittrich, Il-Keun Kong, Hyung-Goo Kim, Thomas Haaf, Barbara Vona

**Affiliations:** 1Institute of Human Genetics, Julius Maximilians University, 97074 Würzburg, Germany; julia.doll@uni-wuerzburg.de (J.D.); susi_kolb@yahoo.de (S.K.); schnapp-linda@web.de (L.S.); liedtke@biozentrum.uni-wuerzburg.de (D.L.); sabine.knaup@uni-wuerzburg.de (S.K.); michaela.hofrichter@uni-wuerzburg.de (M.A.H.); marcus.dittrich@biozentrum.uni-wuerzburg.de (M.D.); thomas.haaf@uni-wuerzburg.de (T.H.); 2Cellular and Molecular Research Center, Sabzevar University of Medical Sciences, Sabzevar 009851, Iran; rad6790@yahoo.com (A.R.); abolfazl_adli@yahoo.com (A.A.); atefeh.hasanzadeh5@gmail.com (A.H.); 3Tübingen Hearing Research Centre, Department of Otorhinolaryngology, Head and Neck Surgery, Eberhard Karls University, 72076 Tübingen, Germany; 4Max Delbrück Center for Molecular Medicine in the Helmholtz Association, 13125 Berlin, Germany; fruesch@mdc-berlin.de; 5Department of Chemistry, Bacha Khan University, Charsadda, Khyber Pakhtunkhawa 24420, Pakistan; imrangnu@gmail.com; 6Institute of Bioinformatics, Julius Maximilians University, 97074 Würzburg, Germany; tobias.mueller@uni-wuerzburg.de; 7Department of Animal Science, Division of Applied Life Science (BK21plus), Institute of Agriculture and Life Science, Gyeongsang National University, Jinju 52828, Korea; ikong7900@gmail.com; 8Neurological Disorders Research Center, Qatar Biomedical Research Institute, Hamad Bin Khalifa University, Doha 34110, Qatar; hkim@hbku.edu.qa

**Keywords:** *CDC14A*, DFNB32, autosomal recessive hearing loss, exome sequencing, splicing, frameshift, non-sense mediated mRNA decay

## Abstract

*CDC14A* encodes the Cell Division Cycle 14A protein and has been associated with autosomal recessive non-syndromic hearing loss (DFNB32), as well as hearing impairment and infertile male syndrome (HIIMS) since 2016. To date, only nine variants have been associated in patients whose initial symptoms included moderate-to-profound hearing impairment. Exome analysis of Iranian and Pakistani probands who both showed bilateral, sensorineural hearing loss revealed a novel splice site variant (c.1421+2T>C, p.?) that disrupts the splice donor site and a novel frameshift variant (c.1041dup, p.Ser348Glnfs*2) in the gene *CDC14A*, respectively. To evaluate the pathogenicity of both loss-of-function variants, we analyzed the effects of both variants on the RNA-level. The splice variant was characterized using a minigene assay. Altered expression levels due to the c.1041dup variant were assessed using RT-qPCR. In summary, cDNA analysis confirmed that the c.1421+2T>C variant activates a cryptic splice site, resulting in a truncated transcript (c.1414_1421del, p.Val472Leufs*20) and the c.1041dup variant results in a defective transcript that is likely degraded by nonsense-mediated mRNA decay. The present study functionally characterizes two variants and provides further confirmatory evidence that *CDC14A* is associated with a rare form of hereditary hearing loss.

## 1. Introduction

Hearing loss (HL) is a highly heterogeneous disorder and belongs to one of the most common sensory disorders in humans with a prevalence of 1–2 in 1000 newborns. Approximately 120 genes responsible for non-syndromic HL have been identified so far [1,2]. Since 2016, the gene *CDC14A* (OMIM: *603504, ENSG00000079335) has been associated with autosomal recessive non-syndromic deafness-32 (DFNB32, OMIM: #608653) [3]. Two years later, it was recognized as causing hearing impairment and infertile male syndrome (HIIMS) [4]. *CDC14A* is located on chromosome 1p21.2 and encodes an evolutionarily conserved protein tyrosine phosphatase (Cell Division Cycle 14A) that is important for centrosome separation and productive cytokinesis during cell division [5]. It is present in the kinocilia of hair cells, as well as basal bodies and sound-transducing stereocilia of the mouse inner ear [3]. Homozygous *Cdc14a* mutant mice showed postnatal degeneration of hair cells but normal length kinocilia. Additionally, degeneration of seminiferous tubules and spermiation defects resulted in infertile male mice [4]. Alternative splicing of *CDC14A* yields six different transcripts, with the largest protein encoding 623 amino acids [3]. To date, only nine homozygous genetic variants in 10 different families with a Middle Eastern background have been associated with hearing impairment (Table 1). Five of these variants with one additional presumed variant also caused male infertility [3,4]. Patients with biallelic *CDC14A* variants present variable degrees of HL that range from moderate to profound in severity [4].

Exome sequencing and gene mapping approaches revealed a homozygous splice variant in a 22.5 Mb homozygous interval on chromosome 1 in two hearing impaired patients from an Iranian family (family 1). Additionally, a homozygous frameshift variant in a 13.6 Mb homozygous interval on chromosome 1 was identified in two Pakistani patients (family 2). Both families had a consanguineous background and both genetic variants were novel. We subsequently evaluated the functional effects of the two variants by assessing aberrant splicing and abnormal gene expression. Our findings widen the spectrum of clinically relevant *CDC14A* mutations that are associated with hearing impairment and reinforce its role within the auditory system.

## 2. Results

### 2.1. Clinical Presentation

All patients reported congenital, bilateral, sensorineural HL (Figure 1). The affected individuals in family 1 reported severe-to-profound HL (II.2 (asymmetrical HL, age 43), II.8; Figure 1a,c), whereas the affected individuals in family 2 (II.1 (age 29), II.2 (age 27)) showed profound HL (Figure 1b,d). A diagnosis of clinical HL of the older sibling in family 2 (II.2) was secured at 1 year of age after his mother recognized signs of hearing impairment. However, HL was suspected since birth and was the case with his younger sibling. Furthermore, the HL in the proband of family 1 can be described as non-progressive. Information about progression is unavailable from the affected individuals in family 2. There were no complaints of vestibular dysfunction or tinnitus in the affected individuals of family 2. The ophthalmic examination of family 1 was normal apart from mild refractive error. In addition to HL, the affected individuals in family 2 (II.1, II.2) suffer from compound myopic astigmatism. Both affected individuals in family 2 (II.1, II.2) are unmarried and have no children.

### 2.2. Identification of Two Novel CDC14A Variants

Exome sequencing and bioinformatics analysis of 174 HL-associated genes (Appendix A) ensued using the genomic DNA of the probands of each family. The proband from family 1 (II.2) revealed a homozygous *CDC14A* (NCBI Reference Sequence: NM_033312.2) splice variant c.1421+2T>C that abolished the donor splice site in intron 14 out of 15 encoded exons and was consistent with segregation analysis within family 1 (Figure 1a). Both affected individuals II.2 and II.8 were homozygous for the c.1421+2T>C variant, while an unaffected brother (II.3) was wild type (WT). A homozygous frameshift variant c.1041dup (p.Ser348Glnfs*) in exon 11 out of 15 exons was detected in the proband from family 2 (II.1) after exome sequencing and bioinformatics analysis, which also co-segregated within family 2 (Figure 1b). The affected individuals II.1 and II.2 were homozygous for the c.1041dup, while the mother (I.2) and unaffected brother (II.3) were heterozygous. Both variants were absent from all applied population databases. All other variants that were identified by bioinformatics analysis of the in silico gene panel were prioritized as benign or could not explain the HL phenotype. Bioinformatic screening of the gene *CDC14A* in our in-house exome database that includes approximately 330 individuals with HL did not reveal additional potentially pathogenic variants. Copy number variations were excluded in the 174 HL-associated genes.

Homozygosity mapping disclosed a 22.5 Mb (chr1: 89,845,926-112,389,040; all subsequent coordinates reported in GRCh37) homozygous interval on chromosome 1 including the *CDC14A* gene (coordinates chr1: 100,810,598-100,985,833) in the Iranian proband (II.2, family 1). The Pakistani family 2 was genotyped with a genome-wide SNP array and homozygosity mapping, identical by descent (IBD), with linkage analysis revealed 15 regions reaching the maximal logarithm of the odds (LOD) score of 1.927. The longest region (chr1: 88,430,037-102,069,696) of 13.6 Mb spans the *CDC14A* gene.

According to four out of five in silico prediction tools (MaxEntScan, NNSplice, GeneSplicer and Human Splicing Finder; Alamut visual, version 2.10), the homozygous c.1421+2T>C variant is predicted to disrupt the exon 14 to intron 14/15 splice donor site (Figure 2d). That loss would likely mediate exon skipping or the use of an alternative cryptic splice site nearby. The duplication c.1041dup (p.Ser348Glnfs*2) in family 2 leads to a frameshift and the incorporation of a premature stop codon two triplicate bp positions downstream.

### 2.3. Functional Characterization of the Splice Variant c.1421+2T>C

The effect of the splice variant (c.1421+2T>C) in intron 14 was subjected to testing using a minigene assay that included cloning of exon 14, as well as the 5′ and 3′ flanking intronic sequences into an exon trapping vector (Figure 2a). PCR amplification and Sanger sequencing of WT and mutant cDNA revealed two products of different sizes (Figure 2b,c) that is the result of a truncation of eight nucleotides (GTAAGAAG) in the mutated sequence (372 bp) compared to the WT control (380 bp) due to activation of a cryptic splice site in exon 14 (Figure 2d). The 257 bp sequence in the empty vector control appeared as expected. The eight nucleotide deletion leads to a frameshift c.1414_1421del, p.Val472Leufs*20 (NM_033312.2) and the incorporation of a premature stop codon 20 triplicate bp positions downstream (Figure 2e). Assuming that a protein is translated, it is expected that the full protein would be truncated by approximately 21%.

### 2.4. Quantification of Relative Expression Levels for CDC14A

To investigate whether the premature termination codon (PTC) induced by the c.1041dup (p.Ser348Glnfs*2) variant triggers nonsense-mediated mRNA decay (NMD) of *CDC14A* in family 2, we quantified relative *CDC14A* mRNA expression. RT-qPCR was performed utilizing whole blood of homozygous (II.2) and heterozygous (I.2) individuals with two different primer pairs targeting a region at the anterior part of the gene (exon 2–3, NM_033312.2) and a region posterior to the frameshift variant in exon 11 (exon 11–12, NM_033312.2). The RT-qPCR analysis showed that the relative expression levels of *CDC14A* were significantly reduced by approximately 99% relative expression for both targeted regions for the homozygous individual (II.2) relative to WT controls. We could observe a difference of 17%–34% (exon 2–3) and approximately 57% (exon 11–12) relative expression for the heterozygous individual I.2 when compared relative to WT controls (Figure 3a). In principle, the introduction of a PTC directly after p.348 would truncate 44% of the amino acid residues in full length CDC14A (Figure 3b).

Both variants have been submitted to LOVD v 3.0 under the accession IDs 00269609 and 00269610.

## 3. Discussion

The *CDC14A* gene encodes a protein that is a member of the highly conserved dual specificity protein-tyrosine phosphatase family existing in a wide range of organisms from yeast to human [6]. Their ability to dephosphorylate serine, threonine, as well as tyrosine residues of different proteins is required for the regulation of essential signaling pathways and biological processes such as protein–protein interactions, cell-cycle progression or apoptosis [7]. The encoded protein CDC14A is thought to be involved in the conservation of hair cells in mice and is essential both for normal hearing and male fertility in humans according to nine previously described mutations [3,4]. Furthermore, in mice, CDC14A appears to be advantageous for perinatal survival [4]. The paralogue CDC14B may compensate for the loss of some CDC14A functions in vivo, but does not compensate for male infertility and hearing impairment with loss of CDC14A [4].

Interestingly, there is a noticeable accumulation of pathogenic variants that are responsible for a distinct hearing impaired phenotype in exon 11 (NM_033312.2) and variants that are responsible for HIIMS in exon 10, containing the core dual-specificity phosphatase domain (DSPc/PTPc; Figure 4). Patients who have thus far been identified with C-terminal truncating variants have moderate-to-profound hearing impairment and males with apparently normal fertility, whereas patients with truncating variants in exons encoding two protein domains (exon 10 encodes part of the DSPc/PTPc domain and exons 5 and 6 encode the dual-specificity phosphatase domain (DSPn)) have HIIMS that includes moderate-to-profound hearing impairment (Figure 4, Table 1). It is thought that transcripts with truncating mutations in exon 11 probably avoid NMD since a short isoform containing only 11 exons (NM_003672) also exists compared to the longest isoform (NM_0033312.2) that has 15 exons [4].

The congenital, sensorineural HL phenotype in our patients that ranged from severe-to-profound in the Iranian family (family 1) and profound in the Pakistani family (family 2) can be compared to already presented HL patients with mutations in *CDC14A* that showed prelingual severe-to-profound deafness (Table 1) [3]. Most of the families with autosomal recessive non-syndromic hearing loss (ARNSHL) come from the ‘consanguinity belt’ that includes regions of North Africa, the Middle East and India and uncovered unique signatures of variants in the past [8]. All previously described variants in *CDC14A* were found in families originating from this specific region such as Pakistan, Iran, Tunisia and Mauritania and show a comparable ethnicity as the two families (Iranian and Pakistani) presented in this study.

In contrast to some of the already described HIIMS patients that showed signs of progressive moderate-to-profound HL [4], progression was unreported in our families. Since both affected individuals in family 1 are female and the unmarried status of both affected males in family 2, we cannot exclude that the newly identified *CDC14A* variants would also cause male infertility.

The process of RNA splicing is an essential post-transcriptional mechanism ensuring the correct junction of neighboring exons by the removal of intermediate intronic sequences through the spliceosome complex [9]. Sequence variants in regions critical for correct splicing can consequently lead to exon-skipping or the activation of cryptic splice sites and are responsible for a variety of different diseases [10]. Cryptic splice sites are normally repressed by stronger splice sites that are located nearby, competing for selection by the splicing machinery [11]. One possible consequence of the splice variant c.1421+2T>C in family 1 that is located in intron 14 was exon skipping of exon 14 due to a disruption and consequently weakening of the splice donor site. We verified the activation of an alternative cryptic splice site located eight base pairs upstream within exon 14 (Figure 2d) that results in a frameshift and a truncated protein (p.Val472Leufs*20; Figure 2e) instead of exon skipping and postulate that this variant is responsible for the hearing impaired phenotype within this family.

The vital process of eukaryotic gene expression such as transcription, translation and degradation of mRNA and proteins is essential to ensure a functional gene product [12]. There are several different pathways that are involved in the decay of defectively transcribed gene products most often starting with poly(A) shortening by the Ccr4-Not deadenylase complex followed by either 5′ end decapping (Dcp1/2) and Xrn1 mediated decay [13] or 3′ end exonucleolytic decay mediated by an exosome [14]. Frameshift or nonsense mutations are often responsible for NMD, a crucial process that prevents the production of truncated or potentially toxic dominant-negative proteins [15]. The c.1041dup frameshift variant in family 2 is responsible for the incorporation of a premature stop-codon likely resulting in a truncated transcript. If translated, over >40% of the transcript would be missing that is likely degraded by NMD. We could confirm significantly decreased relative *CDC14A* expression levels for both the unaffected heterozygous and the affected homozygous individual including both the anterior and the posterior part of the mRNA transcript (Figure 3a). Since we only investigated total mRNA levels isolated from whole blood, targeting distinct transcripts of the *CDC14A* gene, we cannot rule out the possibility of compensatory mechanisms such as upregulation of different gene transcripts in other human tissues [16]. Nevertheless, we have experimentally shown that the truncated *CDC14A* product by the c.1041dup frameshift variant is likely targeted by the mechanism of NMD resulting in insufficient levels of functional transcript for the homozygous proband (Figure 3).

Both variants are anticipated to produce a loss-of-function, either through possible truncated protein from the c.1421+2T>C variant, or NMD that is anticipated to occur as a result of the c.1041dup variant.

In order to distinguish between a regular stop codon and a PTC, the NMD machinery recognizes the position of the PTC within newly produced mRNAs. If the PTC is located at least 50–55 nucleotides upstream of the 3′-most exon–exon junction, degradation by the NMD machinery is likely induced to avoid unfavorable transcripts [17]. Since the PTC of the *CDC14A* frameshift variant c.1041dup is located in exon 11 and is followed by a 3′ exon–exon junction more than 55 nucleotides downstream, NMD is likely triggered. Our results showing reduced relative expression levels for *CDC14A* in patients also confirmed this hypothesis. The PTC that originates from the aberrantly spliced transcript (p.Val472Leufs*20) due to the c.1421+2T>C variant is located in the last exon (exon 15) of *CDC14A*. Since there is no 3′ exon–exon junction further downstream of the PTC, a transcript is likely produced to subsequently escape the NMD machinery.

## 4. Materials and Methods 

### 4.1. Patient Recruitment and Clinical Assessment

Informed written consent was obtained from the families. This study was performed under the tenets of the Declaration of Helsinki and was approved by the Ethics Commission of the University of Würzburg (46/15, approval date: 31 March 2015).

We recruited an Iranian (family 1) and Pakistani (family 2) family who both showed a history of parental consanguinity for a total of four affected individuals, as well as unaffected parents and siblings (Figure 1a,b). The affected individuals (family 1: II.2, II.8; family 2: II.1, II.2) underwent audiological assessment and were tested by pure-tone audiometry adhering to recommendations described in Mazzoli et al. (2003) [18]. Ophthalmic examination of the proband in family 1, as well as both affected individuals from family 2 was performed.

### 4.2. Exome Sequencing

Genomic DNA (gDNA) from the two affected individuals was extracted from whole blood. Diagnostic screening of *GJB2*, the most frequently implicated gene in non-syndromic hearing loss, excluded putative pathogenic variants. gDNA of the probands were exome sequenced. Exome library preparation was performed with the TruSeq Exome Enrichment (family 1) and the Nextera Rapid Capture Exome (family 2) kits (Illumina, San Diego, CA, USA) according to manufacturer’s instructions and paired-end sequenced (2 × 76 bp) with the NextSeq500 sequencer (Illumina, San Diego, CA, USA). A v2 high output reagent kit (Illumina) was used and the data were aligned to the human reference genome GRCh37 (hg19).

### 4.3. In Silico Variant Analysis

Data analysis was performed with GensearchNGS software (PhenoSystems SA, Wallonia, Belgium) and an in-house bioinformatics pipeline. Variant filtering was done with a minor allele frequency <0.01 and alternate allele present at >20%. The pipeline data were analyzed based on the GATK toolkit [19] and BWA based read alignment to the human genome (hg19) following GATK best practice recommendations [20,21]. Quality filtering was performed based on the VQSLOD score. Data from the Greater Middle East Variome Project [22], the Iranome Database [23] and gnomAD [24] were used for population-specific filtering. Variant analysis was done with the use of PolyPhen-2 [25], SIFT [26] and MutationTaster [27], as well as variant information annotated in the Deafness Variation Database (DVD) [28]. Exome CNV analysis was performed using the eXome Hidden MarkovModel (XHMM, version 1.0) approach [29]. Splice-site variants were classified on the basis of in-silico splice predictors such as SpliceSiteFinder-like [30], MaxEntScan [31], NNSPLICE [32], Genesplicer [33] and Human Splicing Finder [34]. Homozygosity mapping was done and visualized with HomozygosityMapper [35]. Mutalyzer was used to assess the effect of the variants on the protein in silico [36].

### 4.4. Gene Mapping Approaches

The exome data of the proband from family 1 was subjected to homozygosity mapping that was performed using HomozygosityMapper [35]. The gDNA from individuals (I.2, II.1, II.2, II.3) from family 2 were subjected to genome-wide genotyping using the Infinium HumanCore-24 v1.0 Bead Chip (Illumina) using manufacturer’s protocols.

Data conversion to linkage format files and quality control (QC) was managed with ALOHOMORA software [37]. Up to 258,000 biallelic SNPs after QC were used for linkage analysis with Merlin [38]. Linkage analysis was done with a recessive genetic model with complete penetrance, a rare disease allele frequency of 0.001 and a pedigree with a consanguinity loop (cousin marriage of parents I.1 and I.2). As genetic coordinates, we used the physical position (1 cM ⇔ 1 Mb, hg19). Linkage regions where the LOD score reached the maximal value of 1.927 for this family, indicate homozygous (autozygous) stretches for the two affected individuals (II.1 and II.2) and where the unaffected brother (II.3) is not homozygous. Linkage analysis was repeated with a linkage disequilibrium (LD) reduced marker set to verify that LOD score peaks are not inflated by LD.

Genomic coordinates are reported using the GRCh37 human genome assembly.

### 4.5. Validation of The CDC14A Variants and Segregation Analysis

Validation of the *CDC14A* frameshift and splice variants was carried out by Sanger sequencing from PCR-amplified gDNA from the probands and available family members using standard cycling conditions and primers (F: 5′-TCCGCAAAGATTAAGTTCATCCC-3′ and R: 5′-TCTGGATCACACTAAGCCAGC-3′) to validate the c.1421+2T>C and (F: 5′-CTGAGGACTTCAGCAGTCAA-3′ and R: 5′-AACTTGGTACTCGTGGCATC-3′) c.1041dup variants (Metabion, Martinsried, Germany). Primers were designed with Primer3 [39]. The amplicons were sequenced with an ABI 3130xl 16-capillary sequencer (Life Technologies, Carlsbad, CA, USA) and the data were analyzed with Gensearch 4.3 software (PhenoSystems SA, Wallonia, Belgium).

### 4.6. Minigene Assay

To test the in silico splice predictions, an in vitro splicing assay was carried out using a modified protocol from Tompson and Young [40]. Briefly, wild-type and mutant *CDC14A* exon 14 (123 bp) were directly PCR amplified from a control individual and the proband with specific primers containing an additional *Xho*I and *Bam*HI restriction site (forward primer with *Xho*I restriction site: 5′-aattctcgagCCGCTGCTGTCATCACTATTA-3′ and reverse primer with *Bam*HI restriction site: 5′-attggatccACCATTCCCTCCACAACCTT-3′). The PCR reaction amplified the entire exon 14 sequence plus an additional 133 bp (5′) and 193 bp (3′) from the flanking intronic regions. After PCR amplification, PCR clean-up, restriction enzyme digestion of the PCR fragments and pSPL3 exon trapping vector was performed prior to cloning of exon A and exon B fragments into the linearized pSPL3-vector and DH5α competent cells (NEB 5-alpha, New England Biolabs, Ipswich, MA, USA. The WT and mutant-containing vector sequences were Sanger sequence confirmed.

Vectors containing either homozygous or WT sequence were transfected into HEK 293T cells (ATCC) at a density of 2 × 10^5^ cells per mL. Of the respective pSPL3 vector 2 µg was transiently transfected using 6 µL of FuGENE 6 Transfection Reagent (Promega, Madison, WI, USA. An empty vector and negative transfection reactions were included as controls. The transfected cells were harvested 24 h after transfection for RNA extraction. Total RNA was prepared using miRNAeasy Mini Kit (Qiagen, Venlo, Netherlands). Approximately 1 µg of RNA was reverse transcribed using a High Capacity RNA-to-cDNA Kit (Applied Biosystems, Waltham, MA, USA) following the manufacturer’s protocols. The cDNA was PCR amplified using vector-specific SD6 forward (5′-TCTGAGTCACCTGGACAACC-3′) and reverse SA2 (5′-ATCTCAGTGGTATTTGTGAGC-3′) primers. The amplified fragments were visualized on a 1% agarose gel and subsequently Sanger sequenced.

### 4.7. Expression Analysis Using Reverse Transcription Quantitative Real-Time PCR (RT-qPCR)

To examine altered expression levels in *CDC14A* due to the loss-of-function variant in family 2, total RNA was extracted from whole blood of probands I.2 and II.2 (family 2) using PAXgene Blood RNA Kit (Qiagen) according to manufacturer’s instructions. cDNA was synthesized using High Capacity RNA-to-cDNA Kit (Applied Biosystems). We performed RT-qPCR using a ViiA7 Real-Time PCR System (Thermo Fisher Scientific, Waltham, MA, USA). Each sample and primer pair was analyzed in triplicates on a single qPCR plate using HOT FIRE Pol Eva Green Mix Plus (Solis BioDyne, Tartu, Estonia). Relative expression levels were calculated using the QuantStudio Real-Time PCS Software v1.3 by ΔΔCt method. Control samples were either used as reference samples or for relative expression comparison. Combined cT values of housekeeping genes *GAPDH*, *IPO8* and *HPRT1* were used for endogenous cDNA controls. Used primer pairs are listed in Table 2.

### 4.8. Statistical Data Collection 

The results of RT-qPCR are represented as relative quantification (RQ) means. Statistical analysis was performed using OriginPro 2018G. Normality (Shapiro–Wilk) and equality of variances (Levene’s) were calculated and a one-way ANOVA with Bonferroni correction for multiple comparisons was subsequently conducted. A *p*-value of less than 0.05 is considered statistically significant.

## Figures and Tables

**Figure 1 ijms-21-00311-f001:**
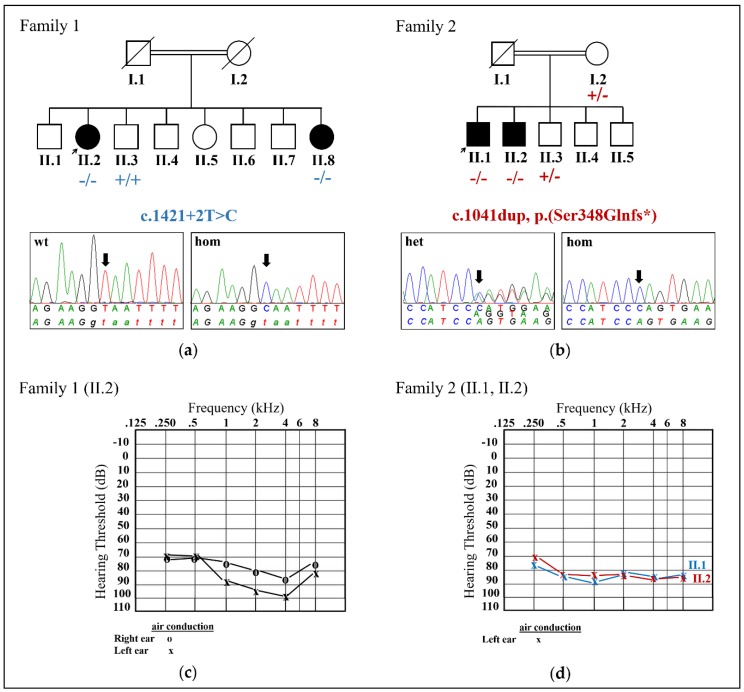
Pedigree, segregation of the *CDC14A* variants in families 1 and 2 and pure-tone audiometry. (**a**,**b**) An Iranian ((**a**) family 1) and Pakistani family ((**b**) family 2) with a consanguineous background, each showing two affected individuals ((**a**), II.2, II.8; (**b**) II.1, II.2) who are shown in black symbols. Unaffected parents and siblings are shown in white symbols. The probands are marked with arrows. The mutated allele is marked with a “-”. The wild type allele is displayed with a “+”. Sanger sequence chromatograms of the *CDC14A* c.1421+2T>C variant in wild type (WT; (**a**), left) and homozygous ((**a**), right) orientation and the *CDC14A* c.1041dup variant in heterozygous ((**b**), left) and homozygous ((**b**), right) orientation. (**c**,**d**) Audiograms showing pure-tone air conduction thresholds of II.2 ((**c**), family 1) and II.1, II.2 ((**d**), family 2). Air conduction thresholds for right and left ears are represented with circles and crosses, respectively. Abbreviations: het, heterozygous; hom, homozygous; wt, wild type.

**Figure 2 ijms-21-00311-f002:**
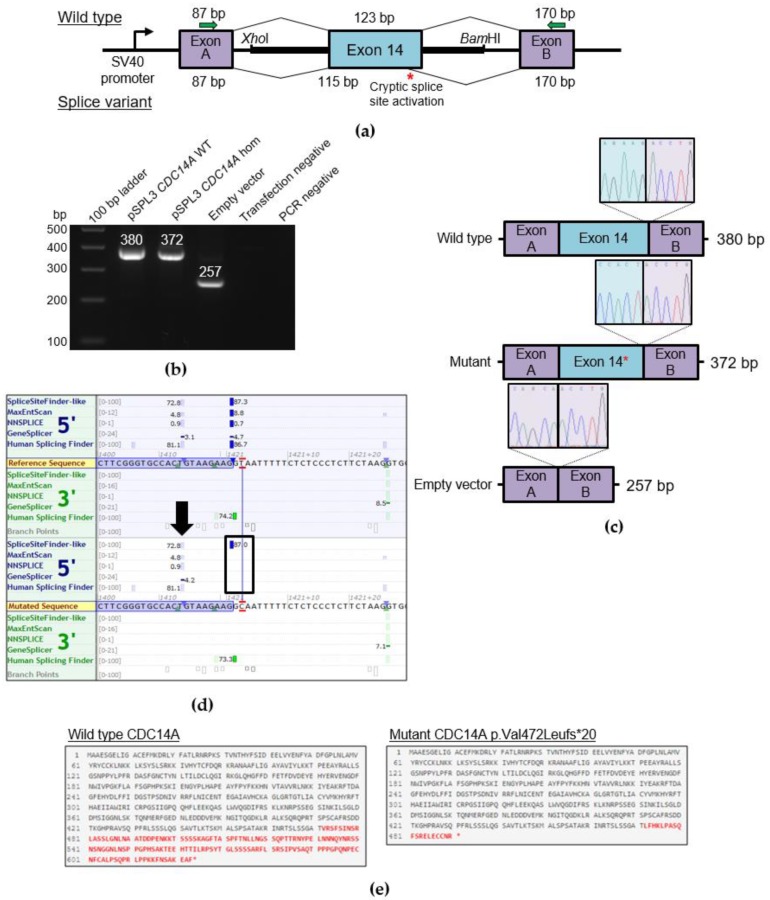
Characterization of the *CDC14A* c.1421+2T>C exchange via a minigene assay. (**a**) A schematic of the pSPL3 exon trapping vector with cloned *CDC14A* exon 14 (blue) and flanking sequence containing *Xho*I and *Bam*HI restriction sites that was directly amplified from proband and wild type genomic DNA. Exons A and B (purple) originate from the vector. A schematic of the resulting splice products is shown, with the wild type splicing profile (top) and splice variant sequence that activates a cryptic splice site (bottom, red asterisk). The PCR primers that were used to amplify the Exon A splice donor region (SD6) and Exon B splice acceptor region (SA2) are depicted by green arrows. (**b**) Electrophoretic visualization of cDNA RT-PCR products amplified from constructs after transfection into HEK293T cells. Amplicons were resolved on a 1% agarose gel. Wild type splicing (lane: pSPL3 *CDC14A* WT) yields a 380 bp product that constitutes the Exon A, exon 14 and Exon B amplified regions. The homozygous mutant amplicon (lane: pSPL3 *CDC14A* hom) shows a band around 380 bp that, when sequenced, indicates a cryptic splice site activation. The empty vector shows the expected 257 bp product. (**c**) Sequencing electropherograms of the exon 14 5′ splice site boundaries for the RT-PCR products for wild type (top), mutant showing cryptic splice site activation (middle) and empty vector (bottom). (**d**) In silico splice prediction tools for the c.1421+2T>C exchange that is marked with red lines visualized with Alamut visual (2.10). The upper panel shows the reference sequence splice scores and the lower panel shows the splice scores for the c.1421+2T>C exchange with multiple in silico prediction tools estimating the loss of the native exon 14 5′ donor splice site that is due to the variant (shown with a black box). In the mutant panel, the splice scores of an adjacent cryptic 5′ donor site are either unchanged or strengthened and marked with a black arrow. (**e**) Effect of the splice variant on the protein, comparing wild type (top) and the truncated (bottom) protein resulting from the aberrantly spliced product (visualized with Mutalyzer). The amino acid residues marked in red are those that are altered due to the variant.

**Figure 3 ijms-21-00311-f003:**
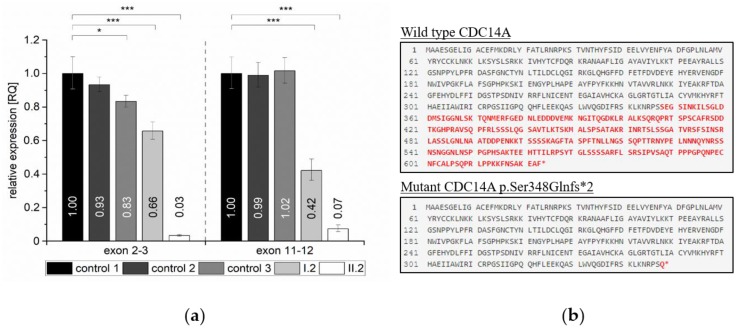
Quantification of the RT-qPCR relative expression values for the *CDC14A* c.1041dup variant in homozygous, heterozygous and wild type control individuals. (**a**) Relative expression levels are shown for exon 2–3 (F_(4, 10)_ = 150.69) and exon 11–12 (F_(4, 10)_ = 112.84) for the normalized reference samples (wt controls 1, 2 and 3), and both tested individuals in family 2 (I.2, heterozygous for c.1041dup and II.2, homozygous for c.1041dup). *N* = 3 for each group. Values are represented as means and error bars extend to the respective minimal and maximal values. To improve readability, significant differences are only indicated for pairwise comparisons to the normalized reference sample (wt control 1). See Appendix A for extended information. * *p* < 0.05 and *** *p* < 0.001. (**b**) Effect of the c.1041dup variant on the protein, comparing wild type (top) and the truncated (bottom) protein resulting from the aberrantly spliced product (visualized with Mutalyzer). The amino acid residues marked in red are those that are altered due to the variant.

**Figure 4 ijms-21-00311-f004:**
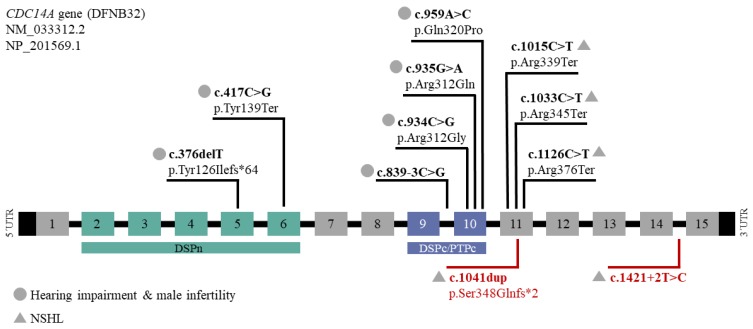
Summary of all variants in the gene *CDC14A* (NM_033312.2, NP_201569.1) that is composed of 15 exons. The dual-specificity phosphatase domain (DSPn) and the core dual-specificity phosphatase domain (DSPc/PTPc) are represented in green and blue bars, respectively. Indicated are the nine previously described mutations (above) and the two newly identified variants in the affected individuals from family 1 and 2 (red, below). The two different phenotypes are indicated with a circle (hearing impairment and male infertility) or triangle (NSHL).

**Table 1 ijms-21-00311-t001:** Summary of previously described and newly identified families with homozygous variants in CDC14A.

Family ID	Variant	Sex	HL Onset	HL Severity	Male Infertility	Ethnicity	Reference
**HLRB11**	c.376delT p.Tyr126Ilefs*64	Male Female	Congenital	moderate-to-profound, progressive	Yes	Pakistani	Imtiaz et al. [4]
**HLAI24**	c.417C>G p.Tyr139Ter	Male Female	Congenital	moderate-to-profound, progressive	Yes	Pakistani	Imtiaz et al. [4]
**FT1**	c.935G>A p.Arg312Gln	Male Female	Congenital	moderate-to-severe, progressive	n.a.	Tunisian	Imtiaz et al. [4]
**HPK1**	c.839-3C>G p.?	Male	Congenital	moderate-to-profound, progressive	Yes	Pakistani	Imtiaz et al. [4]
**MORL1**	c.934C>G p.Arg312Gly	Male Female	Congenital	severe-to-profound, progressive	Yes	Iranian	Imtiaz et al. [4]
**PKDF539**	c.959A>C p.Gln320Pro	Male	Congenital	severe-to-profound, progressive	Yes	Pakistani	Imtiaz et al. [4]
**Mauritanian family**	c.1015C>T p.Arg339Ter	Male	Congenital	Profound	No	Mauritanian	Delmaghani et al. [3]
**PKSN10**	c.1033C>T p.Arg345Ter	Male Female	Congenital	moderate-to-profound, progressive	No	Pakistani	Imtiaz et al. [4]
**MORL2**	c.1126C>T p.Arg376Ter	Male Female	Congenital	moderate-to-profound	No	Iranian	Imtiaz et al. [4]
**Iranian family**	c.1126C>T p.Arg376Ter	Male Female	Congenital	severe-to-profound	No	Iranian	Delmaghani et al. [3]
**Family 1**	c.1421+2T>C p.Val472Leufs*20	Female	Congenital	severe-to-profound	No	Iranian	Present study
**Family 2**	c.1041dup p.Ser348Glnfs*2	Male	Congenital	Profound	n.a.	Pakistani	Present study

Abbreviations: n.a., not available.

**Table 2 ijms-21-00311-t002:** Primer sequences for RT-qPCR.

Exon	5′-3′ Primer Sequence (Forward)	5′-3′ Primer Sequence (Reverse)
*CDC14A* Ex2_3	CCCACTATTTCTCCATCGATGA	GTACACCATTGCCAAGTTCAG
*CDC14A* Ex11_12	TGGCCTAGATGATATGTCTATTG	CTTCTAAGTTATCCTCTCCAAATC
*GAPDH*	TGCACCACCAACTGCTTAGC	GGCATGGACTGTGGTCATGAG
*IPO8*	CGAGCTAGATCTTGCTGGGT	CGCTAATTCAACGGCATTTCTT
*HPRT1*	TGACACTGGCAAAACAATGCA	GGTCCTTTTCACCAGCAAGCT

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
