# Peer review of "Novel Loss-of-Function Variants in CDC14A are Associated with Recessive Sensorineural Hearing Loss in Iranian and Pakistani Patients"

_ijms, 2020, doi:10.3390/ijms21010311_

Round 1

Reviewer 1 Report

The manuscript by Doll and colleagues described two families with ARNSHL. Using a combination of linkage mapping and Exome Sequencing, the authors identified a homozygous predicted deleterious variant in the CDC14A gene in each family. The authors performed functional experiments to show the frameshift variation is a null allele and the splice-altering variant results in a defect in RNA splicing.

Overall, the manuscript is extremely well written, scientifically sound and the conclusions are supported by the results. The manuscript was a real joy to review.

I only have very minor comment:

1. Please be more specific when references figures. For example, line 172-174. These should read Figure 3A and Figure 3B respectively. Please check this throughout the manuscript.

2. The authors should reread the discussion of Imtiaz et al 2018 about transcripts and phenotype association and update the discussion accordingly.

3. While NOT required, if the authors could get a semen analysis on the affected males in family 2, it would confirm these patients are truly DFNB32/105 patients and not HIIMs patients.

4. In figure 4 please update the circle labeling to say “NSHL”

Author Response

The response to Reviewer 1 can be found below:

1. References to the subfigures within each figure were added throughout the manuscript.

2. This information has been added.

3. We thank the reviewer for this suggestion. During an earlier stage of this study, we attempted to get a sample from one of the affected males in family 2 and were unfortunately unsuccessful.

4. We relabelled the deafness (triangle) symbol to say “NSHL”

Reviewer 2 Report

In this manuscript, the authors identify two novel mutations in CDC14A associated with non-syndromic hereditary hearing loss by exome sequencing. To date, only nine mutations have been described in this gene. This study provides more confirmatory evidence that this gene is associated with hereditary hearing impairment. They have performed functional assays to demonstrate the pathogenicity of the variants.

The manuscript is quite complete and I have only a few considerations:

Major revisions:

  1. Authors suggest that c.1421+2T>C variant likely escape the NMD (lines 264-265). Then, they should analyze the effect of this variant using RNA obtained from the patient´s blood.
  2. I wonder if CDC14A is expressed in other samples like saliva or nasal epithelium. It is known that nasal epithelium is a tissue more similar to the inner ear than blood. Could you analyze the effect of the mutations using saliva or nasal epithelium?

Minor revisions:

3. Line 40. Maybe, frameshift word is not very correct, it is more appropriate if you are talking about protein and authors are referring to DNA.

4. c.1421+2T>C involves exon 14. There are several transcripts of CDC14A lacking exon 14 and 15. For example NM_033313. Which transcripts are expressed in the inner ear?

5. Line 239. Authors report that >40% of the transcript is missing by NMD, although they indicate that the expression level of CDC14A is reduced by approximately 99% for the homozygous individual (line 169). Please clarify this point.

6. Line 242-245. Authors indicated “ These expression results imply that CDC14A is not haploinsufficient…..” However, haploinsufficiency is not the most obvious pathogenic mechanism associated with recessive mutations. Then, this phrase is not necessary.

7. Table 1. It would be helpful if mutations were ordered according to the DNA position. The mutation c.1126C>T, identified in two families, could be put together.

8. It is was very interesting to follow the affected male patient in order to know the effect of the variant on male fertility.

Author Response

The response to Reviewer 2 can be found below:

1. We agree that analysis of RNA directly from blood would be informative functionally assess NMD in transcripts expressed in the blood. However, our hypothesis about this splicing variant is based on the contextual knowledge that aberrant splicing occurring in the last exon without a further 3’ exon-exon junction downstream of the PTC. Therefore, the transcript would likely escape NMD machinery (based upon functional data from Maquat et al., 2004). Unfortunately, there is no further opportunity for RNA testing from blood. 

2. Expression data from gtexportal suggest that CDC14A would be expressed in epithelial tissue (unfortunately, nasal epithelium was not distinguished).

https://www.gtexportal.org/home/gene/CDC14A#gene-transcript-browser-block

Only blood was available from Family 2 for expression studies.

3. The word frameshift was removed.

4. Previous publications have focused on the CDC14A human transcript NM_033312.2, so the full function of all human CDC14A inner ear transcripts has not been presently discerned.

Recent mouse transcriptomics data from the mouse inner ear by Ranum et al., 2019 (PMID: 30865901) may provide some insight into this question (please refer to the figure below), especially since this was the first study to use Nanopore sequencing. Based on these mouse transcriptomics data, there is evidence that suggests transcripts with 3’ exons are present in inner ear transcripts and there are only two differences that are marked with small boxes that show exons 5 and 6 are absent in one of the two transcripts identified in the mouse inner ear. By no means is our understanding of the relevant transcripts complete in human and our understanding of the transcripts of the mouse inner ear also appears to be limited. However, at least for the relevant part of the transcript (the 3’ region), it appears that all exons that were originally appearing in the Ensembl transcripts (at least those that are depicted toward the top of the figure) are shared in common.

Figure. Transcriptomics data from the mouse inner ear showing splicing profiles from two transcripts that correspond to Cdc14a-201 and Cdc14a-202. The upper panel shows these two transcripts with exons labelled. The lower panel shows Sashimi plots of transcripts from the inner hair cells, Deiter cells and outer hair cells. Source: morlscrnaseq.org

5. Lines 250-253: This has been clarified and broken into two sentences. If the transcript were translated, over 40% of the amino acid residues would be missing. It is likely this would trigger NMD that is reflected by the approximately 99% reduction of expression.

6. The sentences discussing haploinsufficiency have been removed.

7. The variants have been ordered according to DNA position, as suggested. Due to the subtle phenotype differences in the patients with the c.1126C>T variant, we decided to keep these as two entries but organized these adjacent to one another.

8. Unfortunately, we could not obtain a sample to perform semen analysis. However, we would like to follow-up on the two male patients to assess possible impact on fertility.

Round 2

Reviewer 2 Report

Authors have improved the manuscript.I think it is ready to be published.